# Funneling modulatory peptide design with generative models: Discovery and characterization of disruptors of calcineurin protein-protein interactions

Jérôme Tubiana[1☯], Lucia Adriana-Lifshits[2☯], Michael Nissan[1], Matan Gabay[2], Inbal Sher[2], Marina Sova[2], Haim J. Wolfson[1]*, Maayan Gal[2]*

1 Blavatnik School of Computer Science, Tel Aviv University, Tel Aviv, Israel, 2 Department of Oral Biology, The Goldschleger School of Dental Medicine, Faculty of Medicine, Tel Aviv University, Tel Aviv, Israel

☯ These authors contributed equally to this work.
* wolfson@tau.ac.il (HJW); mayyanga@tauex.tau.ac.il (MG)

## Abstract

Design of peptide binders is an attractive strategy for targeting "undruggable" protein-protein interfaces. Current design protocols rely on the extraction of an initial sequence from one known protein interactor of the target protein, followed by in-silico or in-vitro mutagenesis-based optimization of its binding affinity. Wet lab protocols can explore only a minor portion of the vast sequence space and cannot efficiently screen for other desirable properties such as high specificity and low toxicity, while in-silico design requires intensive computational resources and often relies on simplified binding models. Yet, for a multivalent protein target, dozens to hundreds of natural protein partners already exist in the cellular environment. Here, we describe a peptide design protocol that harnesses this diversity via a machine learning generative model. After identifying putative natural binding fragments by literature and homology search, a compositional Restricted Boltzmann Machine is trained and sampled to yield hundreds of diverse candidate peptides. The latter are further filtered via flexible molecular docking and an in-vitro microchip-based binding assay. We validate and test our protocol on calcineurin, a calcium-dependent protein phosphatase involved in various cellular pathways in health and disease. In a single screening round, we identified multiple 16-length peptides with up to six mutations from their closest natural sequence that successfully interfere with the binding of calcineurin to its substrates. In summary, integrating protein interaction and sequence databases, generative modeling, molecular docking and interaction assays enables the discovery of novel protein-protein interaction modulators.

## Author summary

Peptides that efficiently bind a target protein and interfere with its native protein-protein interactions are attractive reagents for basic research and therapeutic applications. However, rational peptide design remains challenging, as i) exhaustive exploration of the vast

**Data Availability Statement:** The following are made available as supplementary data: The list of seed Cn interactors (Supplementary Data 1), the

multiple fragment alignment (Supplementary Data 2), the list of 768 peptides analyzed by docking and microarray binding (Supplementary Data 3), the set of sequence motifs learnt by the model (Supplementary Data 4). Source code for training Restricted Boltzmann Machines are available from https://github.com/jertubiana/PGM; the PepCrawler flexible docking algorithm is available as a webserver from http://bioinfo3d.cs.tau.ac.il/PepCrawler/ and as a Linux executable with Python 3 API from http://bioinfo3d.cs.tau.ac.il/PepCrawler/download/PepCrawler.zip (Supplementary Data 5).

**Funding:** This work was supported by the Zimin Institute for Engineering Solutions Advancing Better Lives (H.J.W and M.G.), the TAD Center for Artificial Intelligence & Data Science (H.J. W. And M.G.) the Edmond J. Safra Center for Bioinformatics at Tel Aviv University (J.T.), the Human Frontier Science Program (cross-disciplinary postdoctoral fellowship LT001058/2019-C, J.T.), the ADAMA Center for Novel Delivery Systems in Crop Protection (Ph.D. scholarship, L. A-L.), Len Blavatnik and the Blavatnik Family Foundation (H.J.W.). The funders had no role in study design, data collection and analysis, decision to publish, or preparation of the manuscript.

**Competing interests:** The authors declare no competing interests.

sequence space is impossible, ii) generically, there is a mismatch between selection criteria and target objectives, and iii) additional constraints such as low toxicity are frequently critical. Here, we present an integrative peptide design protocol based on a sequence generative model trained on native protein interactors of the target. We tested our protocol on Calcineurin, a serine/threonine phosphatase involved in multiple health and disease pathways. We showed that the generative model i) enables extensive exploration of the sequence space, ii) approximates well binding affinity to the target, and iii) yields highly diverse candidate sequences. After further selection via molecular docking and high-throughput binding assay, we found that 70% of the designed peptides successfully interfered with Cn-substrate interactions. Our integrative protocol could thus be broadly applicable to the rational design of protein-protein interaction disruptors.

## Introduction

Protein-protein interactions (PPIs) are essential components of all cell signaling pathways [1]. As such, chemical and biological modulators capable of interfering with specific PPI networks are of great importance for fundamental and applied research. However, the design of PPI inhibitors, especially in the form of small molecules, remains a major challenge, mainly due to the physico-chemical properties of protein-protein interfaces. The latter are typically larger, flatter and more flexible than their counterpart enzymatic active sites [2]. These factors limit the inhibitory potential of small molecules and the accuracy of computational molecular docking tools—which heavily rely on shape complementarity [3,4]. Peptides, *i.e.* proteins of small length (L<30–40) with no stable fold, are a promising class of PPI perturbators. They are easy to synthesize and can interfere with native PPI by mimicking the binding site of one of the partners. Their potential coverage is high, as it is estimated that up to 40% of human PPIs involve at least one disordered, peptide-like binding region—particularly in cell signaling and regulatory pathways [5]. Although peptides have limited direct therapeutic applications due to their limited oral bioavailability and fast degradation rate *in-vivo* [6], designing high-affinity peptides can i) provide structural insights into transient PPIs that are challenging to characterize experimentally, ii) suggest pharmacophore hypotheses for *in-silico* screening of small molecules [6] iii) facilitate small molecule screening based on *in-vitro* competition assay (e.g. by Fluorescence Polarization [7–9]) and iv) lead to peptidomimetics-based therapeutics [6,10,11].

The main challenge of peptide discovery lies in the exhaustive and accurate exploration of the sequence space, as there are $20^L$ peptides of length L. For L>10, this is well beyond the capabilities of experimental investigation and computational approaches based on molecular docking. Nonetheless, a crucial edge of inhibitory peptide discovery is that protein fragments that bind the target protein already exist in nature. This observation has laid the basis for current peptide discovery protocols: starting from a known protein-protein complex structure, an initial peptide sequence is derived from the binding interface of one partner [11], and its binding affinity is subsequently optimized by *in-silico* [12–21] or *in-vitro* [22–25] mutagenesis. Such protocols typically explore only a local neighborhood of the sequence space, and cannot readily screen for additional desirable properties such as high binding specificity, high solubility or low immunogenicity.

Recent advances in machine learning sequence generative models (SGM) have proven highly successful at i) learning the biophysical constraints underpinning the functionality of native proteins from raw sequence data and ii) rapidly exploring the sequence space towards the design of artificial proteins with native-like functionality [26–31]. However, training

accurate SGM necessitates a large and diverse set of evolutionary-related sequences with similar functionality [28,32]. Directly transposing this methodology to peptide design is challenging because although additional binding fragments could also a priori be obtained by homology search, the target PPI i) may only be conserved in a few eukaryotic organisms and/or ii) may be mediated by highly conserved short linear motifs (SLiMs). Thus, SGM-guided peptide design has been limited to cases where diverse sequence datasets are available, such as for antimicrobial, anticancer or cell-penetrating peptides [33–36]. Yet, in many PPIs, at least one of the partners is highly multivalent, *i.e.*, it interacts with multiple protein interactors, and the corresponding binding regions are highly overlapping. This provides the opportunity to learn from diverse sequence fragments that are evolutionary-unrelated but have similar binding functionality.

One important caveat of learning from natural partners is that many interact only transiently with the target, with low binding affinities in the $10^2$–$10^3$ micromolar range. Therefore, additional in-silico and/or in-vitro screening for filtering high-affinity peptide binders must complement the SGM. All these considerations motivate us to build an integrative peptide design protocol consisting of four steps: (i) Construction of multiple alignments of putatively binding fragments extracted from known and presumed binders (ii) Training and validation of a SGM, and generation of a library of candidate peptide sequences (iii) Filtering of the library by in-silico flexible protein-peptide docking and in-vitro microarray chip binding assay.

Herein, we implement and evaluate our peptide design protocol on the PPI between Calcineurin (Cn), a calcium-dependent protein phosphatase and its substrates containing the conserved SLIM PxIxIT. Among these is the Nuclear Factor of Activated T-cells (NFAT), a transcription factor playing a key role in a broad range of normal cellular processes and in several diseases [37–39]. Although clinically approved inhibitors of Cn exist (Cyclosporine A and Tacrolimus), they obstruct the Cn catalytic site, inhibiting its activity across all substrates and leading to undesirable side effects such as nephrotoxicity and hepatotoxicity [40,41]. Peptide-based modulators interfering with the binding of Cn to its substrates while keeping its catalytic site available could therefore be favorable alternatives [22,41,42]. Importantly, Cn is highly multivalent [43], and its signaling network has been extensively studied in yeast and humans [44,45]. Reliable information about its various protein substrates and their corresponding binding sites are therefore available, thus facilitating our case study.

## Results

### The Calcineurin signaling network relies on the PxIxIT and LxVP SLIMs

Calcineurin (Cn) is a heterodimeric calcium-dependent phosphatase conserved in metazoans, constituted by a catalytic (~510 amino acids) and a regulatory domain (~170 amino acids); its structure is shown in **Fig 1A**. Upon calcium chelation and interaction with calmodulin (both mediated by the regulatory domain), Cn adopts its active conformation (depicted here) in which its catalytic site (shown in blue) and binding regions are exposed [43,46]. In turn, Cn substrates—most of which are intrinsically disordered—bind it, enabling dephosphorylation of serine and threonine residues by Cn. The NFAT family—a set of five transcription factors conserved in vertebrates—are known examples of Cn substrates; upon dephosphorylation by Cn, they undergo conformational changes that expose nucleus localization motifs, allowing translocation to the nucleus and in turn, binding to DNA. More generally, the Cn signaling network was systematically investigated in mammals and yeast using combinations of *in-vivo*, *in-vitro* and *in-silico* methods, and at least 29 and 38 protein substrates were identified with high confidence respectively for human and yeast [44,45]. To determine the regions tethering

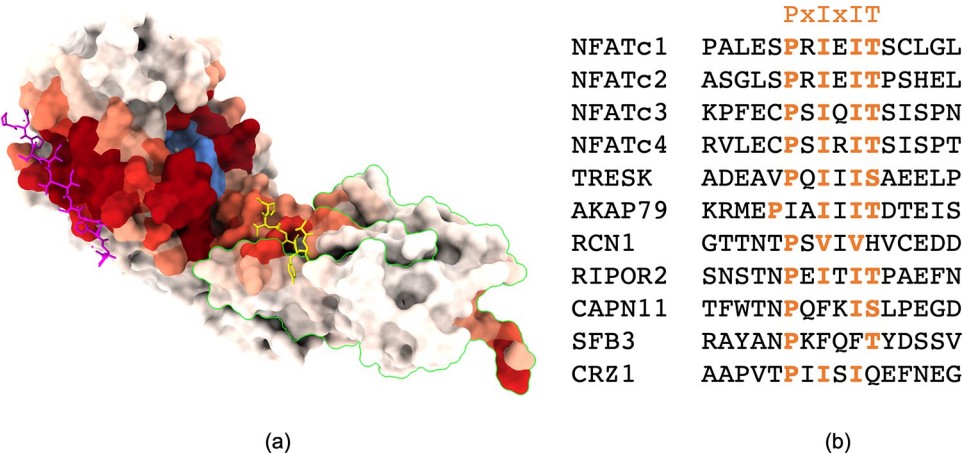

(a)                                                   (b)

**Fig 1. Overview of calcineurin-substrate complexes.** (a) Structure of calcineurin bound to representative SLiM-containing peptides (pdb codes: 2p6b [51], 5sve [52]). calcineurin is shown in molecular surface representation, and is colored by propensity to bind disordered regions (from white = low to dark red = high), based on ScanNet [47,48]. The catalytic site is colored in blue. Both the catalytic and the regulatory (circled in green) domains are shown. PxIxIT and LxVP-containing peptides are shown in stick representation (resp. in magenta and yellow). (b) Sequence alignment of the PxIxIT short linear motifs that bind calcineurin.

the substrates, we used the ScanNet web server [47,48] to predict binding sites of intrinsically disordered proteins (shown as red scale coloring in **Fig 1A**). In addition to the catalytic site, two substrate binding sites are found. Previous studies showed that they recognize two SLiMs: PxIxIT and LxVP, where uppercase letters stand for conserved residues and x represents alternate amino acids. Both motifs i) bind Cn in isolation (crystal structures of representative Cn-bound PxIxIT and LxVP motifs are depicted in respectively magenta and yellow of **Fig 1A**) and ii) are conserved across a wide range of substrates as illustrated for the NFAT isoforms in **Fig 1B**. Substrate-derived, PxIxIT-containing fragments bind relatively weakly to Cn, with dissociation constants $k_d$ ~ 0.5–250 uM [43,49,50]. Indeed, higher affinity interactions may be deleterious *in vivo*: for instance, the Cn-NFAT interaction is evolutionarily tuned to occur only at high calcium concentrations [22]. This pushed the design of PxIxIT peptide variants with higher affinity such as the PVIVIT peptide (kd~ 0.5–2.0 uM) and its peptidomimetics derivatives (up to kd~ 2.5 nM) [22,24,42] that can successfully outcompete Cn-substrate binding in the cell and hence dephosphorylation. However, these peptides were mostly discovered by experimental screening of sequences close to NFAT-derived peptides, without exploring the vast range of additional substrates motifs. Herein, we pursued this strategy and aimed to design alternative peptides capable of competing with the known PVIVIT peptide. New peptide sequences pave the way for further design of variants with higher affinity, specificity and/or solubility than previous sequences.

## An integrative peptide design protocol

The main steps of the proposed design protocol are illustrated in **Fig 2** and summarized hereafter; see Methods for additional details.

**Step 0: Curation of known Calcineurin-binding fragments from literature survey.** Our protocol starts from a list of 67 protein substrates of Calcineurin (Cn) from human and yeast that have been previously characterized [44,45], together with their corresponding PxIxIT-containing fragment(s) (**S1 Data**).

**Step 1: Data augmentation by homology search.** Since this number is too limited for meaningful sequence generative modeling, we first enriched this set by performing homology

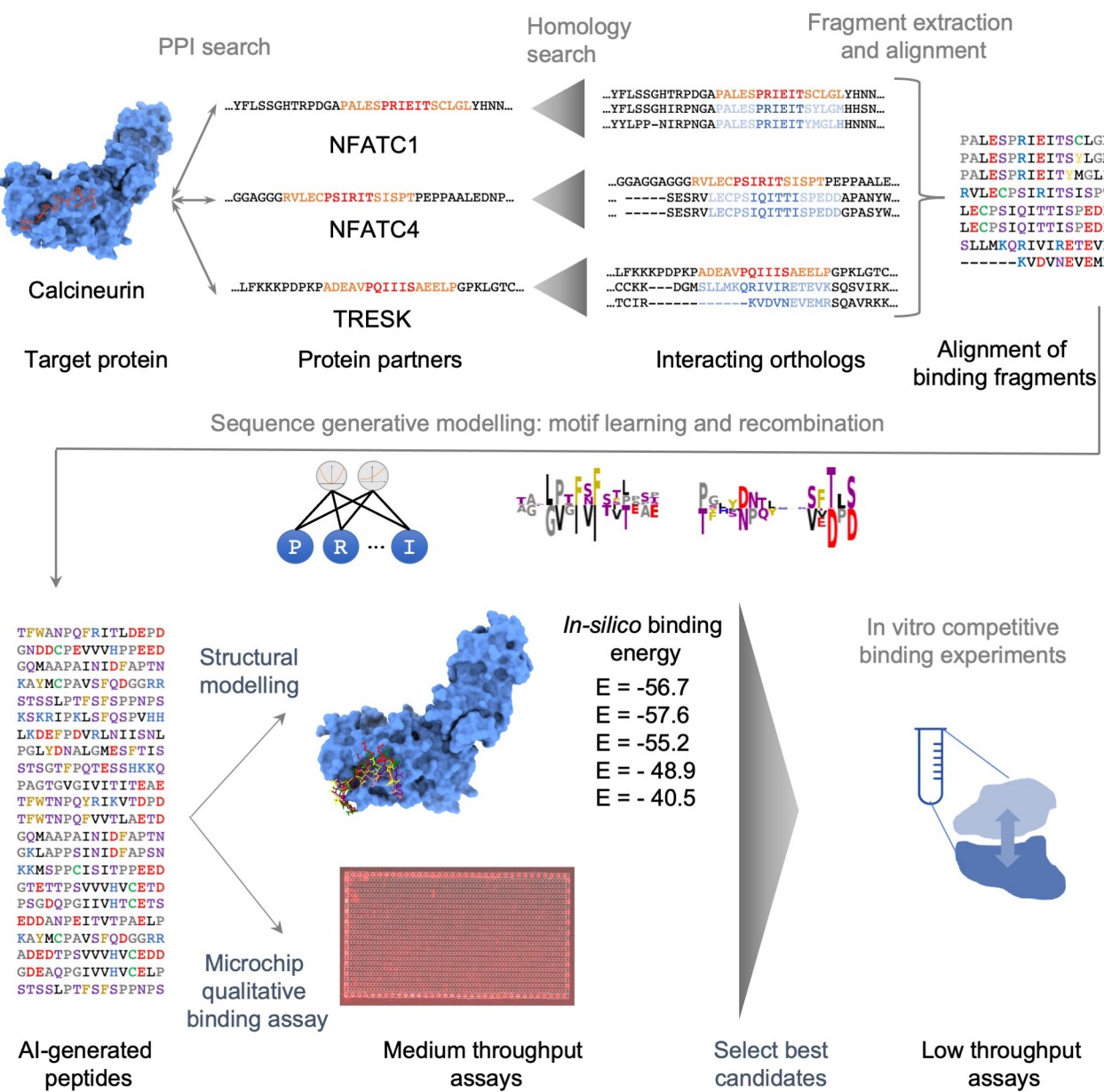

**Fig 2. Overview of the protocol for design of peptide inhibitors of a target PPI.** The protein substrates of the target enzyme are first identified from previous experiments together with their binding fragment. Additional interacting orthologs are identified by homology search, and the corresponding binding regions are extracted and aligned. A sequence generative model (SGM) is trained to generate a library of candidate peptides. The latter are screened for affinity by structural modeling and high-throughput binding assay. The best candidates are selected for further low-throughput experimental characterization.

search across sequence databases to identify additional PxIxIT-like fragments in homologous sequences for each of the listed substrates. Importantly, the protein-protein interaction (PPI) is not guaranteed to be conserved across all orthologs/paralogs, especially in cases like the Cn signaling networks, which undergo rapid rewiring throughout evolution [44]. Therefore, a two-stage sequence-based statistical filtering protocol is applied to eliminate presumed non-interacting homologs. After realignment and deduplication, we obtained a multiple sequence alignment of natural, putatively Cn-binding fragments (**S1 Fig**).

**Step 2: Peptide library design using a Sequence Generative Model.** Next, we trained a compositional Restricted Boltzmann Machine (cRBM), an interpretable sequence generative model inspired by statistical mechanics and previously shown to be suitable for protein sequence modeling [53,54]. After quantitatively and qualitatively validating the model learned (**Figs 3 and S3**), we used it to generate a diverse library of $10^{2-3}$ candidate peptide sequences. We also constructed negative and positive controls for subsequent quality assessment.

**Step 3: In-silico and in-vitro screening.** We then estimated in-silico the binding strength of the various peptides to Cn by template-based docking followed by flexible backbone refinement using Modeller [55] and PepCrawler [56] (**Figs 4 and S4**). In parallel, we performed a medium-throughput qualitative binding assay using a PEPperPRINT peptide microarray to evaluate the direct binding of Cn to selected peptides (**Figs 4 and S5, S3 Data**). The most promising peptides were selected for further characterization.

**Step 4: Quantitative binding assay.** Finally, we experimentally quantify the ability of the designed peptides to compete with the binding of PVIVIT peptide for Cn via Fluorescence Polarization (FP) assay (**Figs 5, S6 and S7**).

## Natural Cn-binding peptides are highly diverse (Steps 0&1)

After the homology search, we obtained a multiple alignment of 1886 fragments and 16 columns, corresponding to the six motif positions and five flanking residues on each side (S2 Data). Sequence logo visualization (**S2A Fig**) revealed high sequence diversity: strong amino acid preferences were only observed at positions P, $I_1$ and $I_2$, respectively for proline and hydrophobic residues. T-SNE projections showed that fragments mostly cluster by source gene and taxon (**S2B and S2C Fig**). Sequence logo visualization of individual gene sub-alignments revealed stronger conservation patterns than for the whole alignment (**S2D Fig**). The central SLIMs are more diverse than PxIxIT, *e.g.* gene-specific SLIMs P[T/S]F[N/S]FS, P[E/D] IT[V/I]T, PQ[F/Y]x[I/L/V]x, P[F/Y][M/V]xFx are found. Conservation patterns are also found outside of the six canonical SLIM positions: *e.g.* for the CAPN11 gene, positions -5,-4, -1, +1, +5 are conserved, suggesting that they are also important for binding. In summary, natural Cn binders have diverse sequences that are not well recapitulated by a single SLIM or PSSM model. Such a combination of local conservation and global diversity may have arisen from multiple binding conformations and/or distinct spatial repartition of the binding energy. Recombining these motifs may yield synthetic sequences with similar or improved binding compared to their natural counterparts. Moreover, it may enable specific competition with a defined substrate, while maintaining binding for others.

## Reverse engineering of binding fragments by Sequence Generative Models (Step 2)

Generative modeling is an unsupervised learning modality. It consists of fitting a parametric probability distribution $P_\Theta(S)$ over the sequence space by maximizing over the parameters $\Theta$ the average log-likelihood $\langle log\, P_\Theta(S) \rangle$ of observed sequences. Since $P_\Theta(S)$ is normalized to unity ($\Sigma_S P_\Theta(S) = 1$), this maximization protocol amounts to assigning large values of $P_\Theta(S)$ for observed sequences and low elsewhere (**Fig 3A**). Thus, the learned $P_\Theta(S)$ qualitatively reflects the evolutionary fitness function, which is also high for evolutionary selected sequences and low for unobserved sequences that were eliminated throughout evolution [32]. Importantly, $P_\Theta(S)$ must be "smooth" in sequence space, as the observed sequences only sparsely samples the set of all evolutionary fit sequences: unobserved but evolutionary fit sequences should also have high probability. After training is completed, novel high-probability sequences distinct from the training data can be generated, and are potential Cn binders. The choice of functional

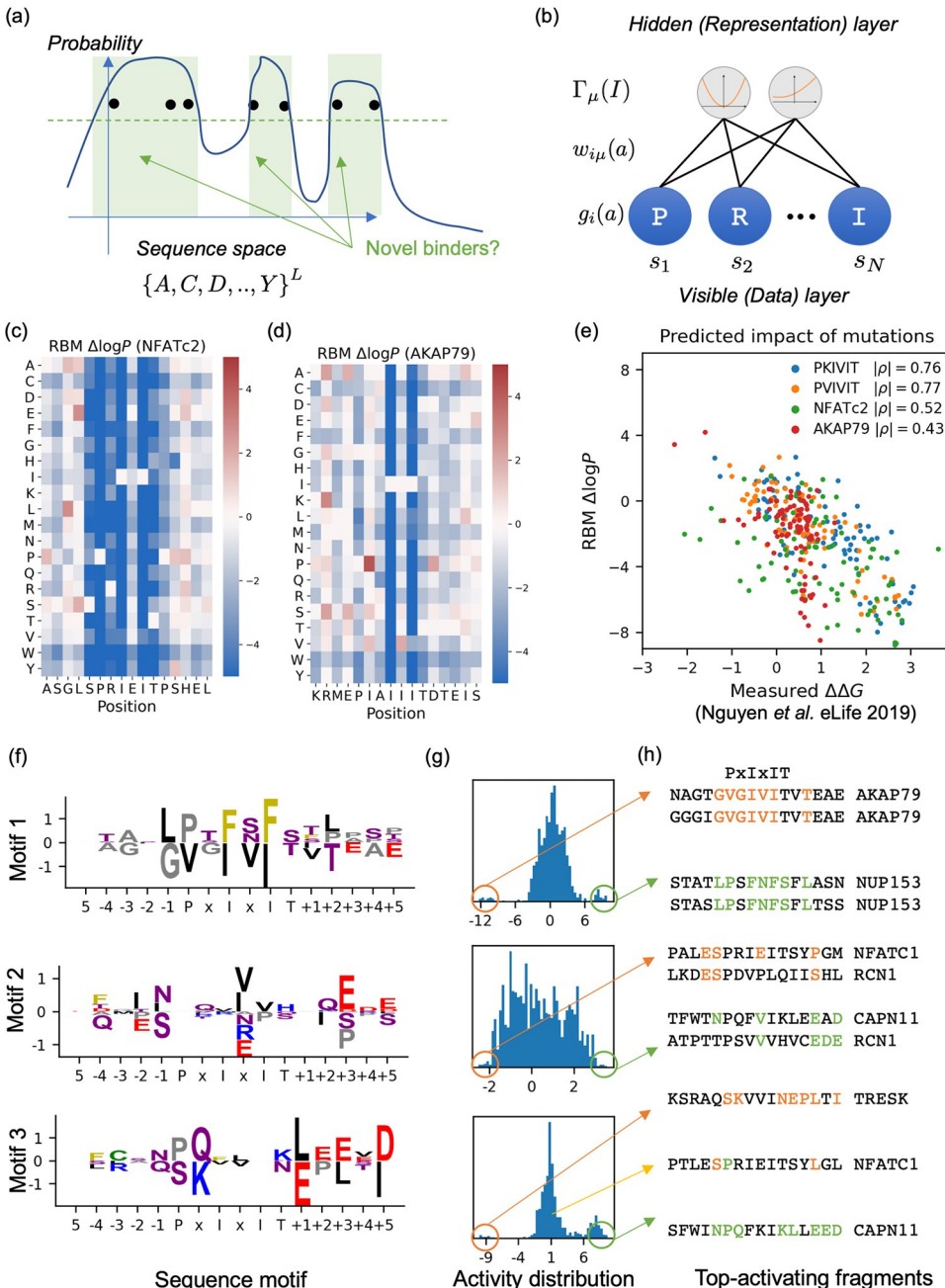

**Fig 3. Generative modeling of PxIxIT binding motifs.** (a) Schematic view of the generative approach. A "smooth" probability distribution over the whole sequence space is learnt from a limited number of samples. Unseen sequences with high probability are potential novel binders, whereas regions with low probability are likely non-functional proteins. (b) Graphical depiction of the cRBM model, the parametric form chosen. The visible layer corresponds to the aligned sequence; each visible unit contributes a site-specific term $g_i(s_i)$ to the log-likelihood. The hidden (representation) layer corresponds to unobserved hidden units, each of which contributes an additional term $\Gamma_\mu(\sum_{i=1}^N w_{i\mu}(s_i))$ to the log-likelihood function, defined as a linear projection through a sparse tensor followed by a trainable, strictly convex non-linearity. (c,d) cRBM-predicted mutational landscapes for the NFATc2 and AKAP79 peptides. Red, white and blue entries correspond respectively to beneficial, neutral and deleterious mutations. (e) Comparison between cRBM-predicted mutational landscapes and deep mutational scans of change in binding affinity measured by Nguyen et al. Four DMS were performed taking as wild type the PVIVIT, PKIVIT, NFATc2 and AKAP79 peptides. Spearman correlation coefficients are annotated. (f,g,h) Selected examples of sequence motifs learnt by the cRBM (f), together with their activity distribution (g) and top-activating sequences (h). Motif 1 is gene-specific, whereas motifs 2 and 3 are shared by multiple genes.

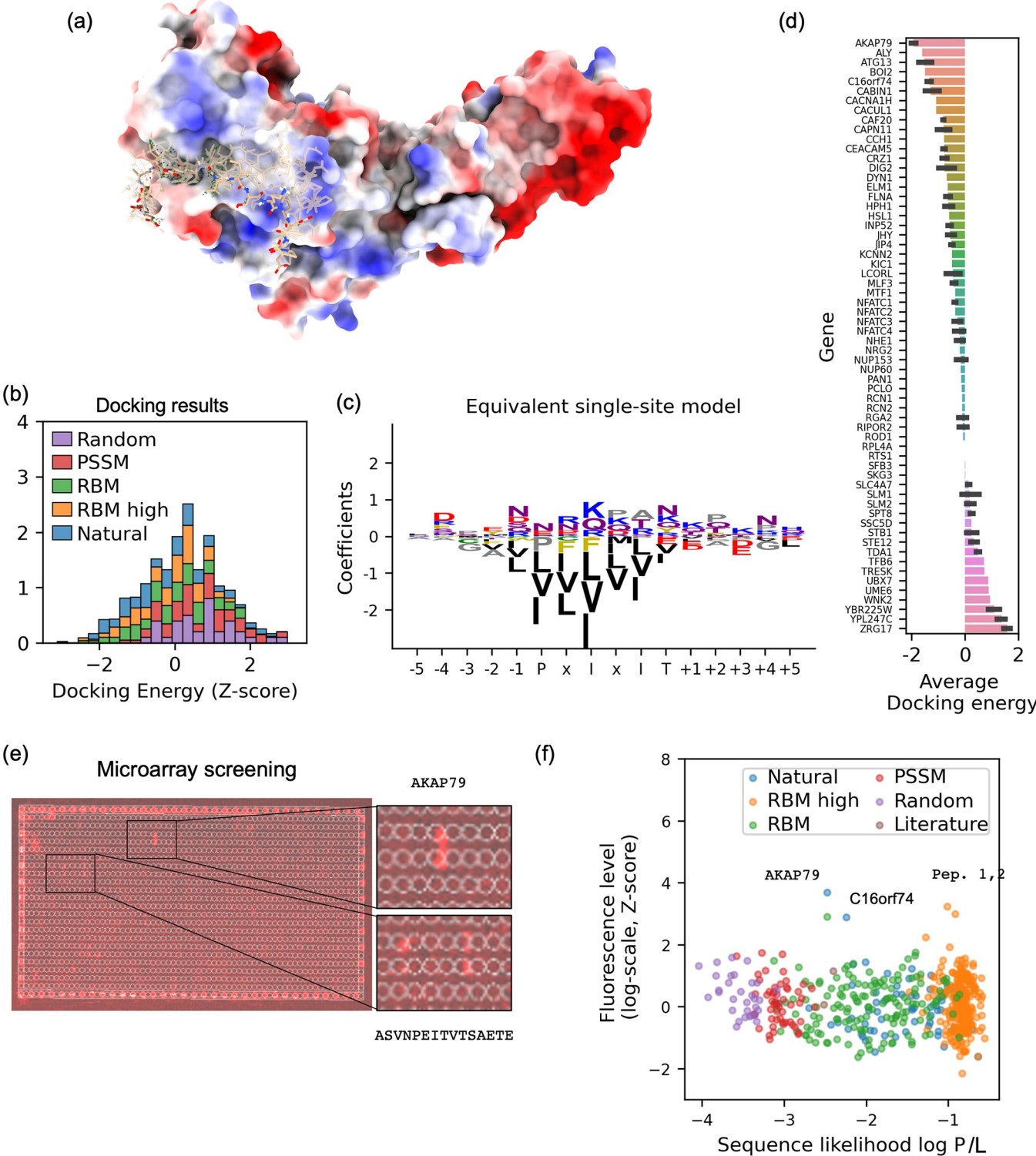

**Fig 4. Medium-throughput filtering by structural modeling and microarray screening.** (a) Depiction of the structural modeling protocol: after alignment to the known PxIxIT binding site, an efficient flexible backbone structure refinement algorithm is applied to estimate the docking energy. (b) Histogram of docking energy scores for the generated peptides and selected controls (lower is better; normalized to zero mean and unit variance). (c) Coefficients of the equivalent single-site model fitted by sparse linear regression, shown in weight logo representation. At each position, the height of the letter is proportional to the corresponding coefficient of the regression; residues with large negative coefficients (e.g. hydrophobic residues at the motif locations) contribute favorably to the docking score. Colors indicate physical property (black = hydrophobic, red = negatively charged, etc.). (d) Per-gene distribution of docking scores across

natural fragments (lower is better). The docking protocol qualitatively discriminates between obligate and transient interactions. (e) Overview of the microarray screening. Peptides are printed on the chip (two circles per peptide). After pouring of Cn and subsequent washing, fluorescent-tagged, a Cn-targeting antibody is overlaid and an image is taken. Fluorescent spots indicate strong Cn binders. (f) Scatter plot of the sequence likelihood (normalized by length, higher is better) against fluorescence level (higher is better, see Methods for details of the data analysis).

form $P_\Theta(S)$ determines the "smoothness" prior (*i.e.*, the inductive bias) over the discrete sequence space. Here, we used the compositional Restricted Boltzmann Machines (cRBM, **Fig 3B**), formally defined as follows. Let $S = \{s_1,..,s_i,..,s_N\}$ be a protein sequence of length $N$, with $s_i \in \{A,C,D,...,Y-\}$ where—is the alignment gap symbol. Its probability $P(S)$ writes:

$$P(S) = \frac{1}{Z} exp[\sum_{i=1}^{N} g_i(s_i) + \sum_{\mu=1}^{M} \Gamma_\mu(\sum_{i=1}^{N} w_{i\mu}(s_i))] \quad (1)$$

Where is a normalizing factor (the partition function) such that $P$ is normalized, $g_i(a)$ are Nx21 site-specific amino acid fields, $w_{i\mu}(a)$ is a Nx21xM sparse weight tensor for projecting the sequence into a continuous, M-dimensional space (termed the hidden unit space) and the potentials $\Gamma_\mu(I)$ are trainable, strictly convex non-linearities such as quadratic functions. Informally, the fields quantify amino acid preferences at each site: high scores are assigned to sequences if their amino acids match the preferred ones at each location. Each weight matrix $w_{.,\mu}(.)$ informally represents a sequence motif consistently found in a subset of the data. The projection $I_\mu(S) = \sum_{i=1}^{N} w_{i\mu}(s_i)$ quantifies the degree of matching between a given sequence and the motif, and the model allocates high probabilities to sequences that have either large positive *or* negative $I_\mu(S)$ via the quadratic-like non-linearity $\Gamma_\mu(I)$. Note that for a quadratic potential $\Gamma_\mu(I) \propto I^2$, Eq (1) reduces to a Potts model with low-rank interaction matrix $J_{ij}(a,b) \propto \sum_\mu w_{i\mu}(a) w_{j\mu}(b)$. Eq (1) is derived from the classical RBM definition (*i.e.* an undirected graphical model with pairwise, bipartite interactions) by marginalizing the joint distribution over the hidden units [53]. After training, novel sequences can be generated by combinatorial recombination of positive and negative motif matches. The cRBM was shown to be a powerful inductive bias for protein sequence modeling, as it generalizes over single-site and pairwise Potts models by incorporating sparse, high-order epistatic interaction terms and

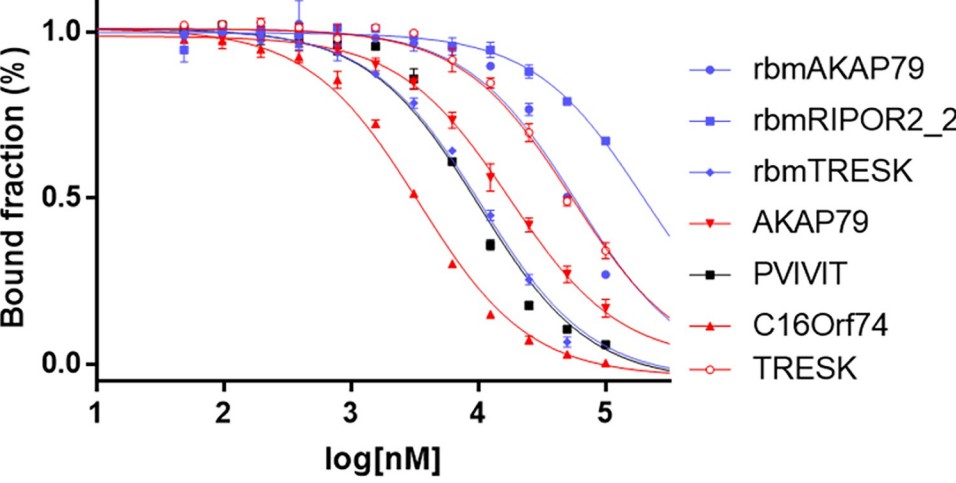

**Fig 5. FP competition assay of selected peptides for the binding of Cn to PVIVIT peptide.** Variable concentrations of each selected peptide were incubated with Cn bound to the FITC-labeled PVIVIT peptide. Polarization levels were read and normalized values were fitted to a single site model. The curve shows the bound fraction of Cn to PVIVIT vs. the logarithmic concentration of the peptides. Red—natural peptides, purple—designed peptides, black—PVIVIT.

is easier to interpret than a pairwise model or deep generative models [53,54]. We trained multiple cRBMs onto the multiple fragment alignment by varying M and regularization strengths, and selected one that best compromised between performance and interpretability (**Methods**, **S3A and S3B Fig**). Here, performance and interpretability of a model were respectively quantified by the log-likelihood of held-out test data (*i.e.* its ability to assign high probability to unseen natural binders) and the fraction of non-zero entries of $w_{i\mu}(a)$ (lower is more interpretable). Sparsity reduces overfitting, eases visualization and interpretation of the learnt sequence motifs, and stirs the generative model to a so-called compositional regime in which only a few motifs are present per sequence [57,53,54].

To validate the model, we compared its learnt log-probability function to deep mutational scans of binding affinity recently performed by Nguyen *et. al* [24]. We computed log-likelihood differences $\Delta log\ P$ for all single-point mutants of four PxIxIT peptides: the natural fragments extracted from the human NFATc2 and AKAP79 proteins, and the synthetic PVIVIT and PKIVIT peptides (**Fig 3C and 3D**). **Fig 3E** shows a scatter plot between experimentally determined changes of binding affinity $\Delta\Delta G$ (lower is better) and predicted fitness change $\Delta log\ P$ (higher is better). An excellent correlation was found for the PKIVIT and PVIVIT peptides (Spearman correlation $-\rho = 0.76, 0.77$ respectively), and a good one for NFATc2 and AKAP79 ($-\rho = 0.52, 0.43$ respectively). This agreement is substantially better than previous structure-based $\Delta\Delta G$ predictions performed using the Rosetta flex_ddG and FoldX ddG protocols [24] ($\rho = 0.2, 0.21$ for PVIVIT and $0.21, 0.39$ for AKAP79). The cRBM-based $\Delta log\ P$ also correlated better with $\Delta\Delta G$ measurements than log-likelihood differences computed using ProteinMPNN [58], a recently published graph neural network for structure-based protein design ($-\rho = 0.53, 0.45, 0.53, 0.36$, reference complex: 2p6b:AB-E), or with a PSSM (*i.e.* independent model) trained on the same alignment (**S1 Table**). While mutations of flanking residues were in general better tolerated than motif residues (**Fig 3C and 3D**), we also found the former to be important as well. To quantify it, we performed additional virtual deep mutational scans for 100 representative natural fragments of the alignment (randomly sampled from the alignment by Kmeans++ algorithm), and calculated the distribution of changes of likelihood upon mutation for each position (**S3C Fig**). Positions that least tolerated mutations on average were, in order, -1, P, $I_1$, +3, $I_2$ and -3. Additionally, visualization of the effective epistatic couplings between pairs of positions (**S3D Fig**) showed important covariation between central and flanking residues (especially -1). This corroborates previous biochemical studies showcasing the importance of flanking residues for Cn binding [24]. Finally, we investigated whether the model was able to identify common motifs shared between different substrates. To this end, we visualized the sequence motifs learnt (three representative motifs shown in **Fig 3F**, the remainder provided as **S3 Data**), together with their matching score distribution (**Fig 3G**) and selected positive and negative top-matching fragments. We found that some motifs, such as motifs 1 (focusing on the central residues) and 3 (focusing on the C-terminal end) were substrate-specific, as evidenced by the trimodal matching score distribution and the fact that its top-activating fragments came from identical substrates. Others, such as motif 2 (simultaneous presence of asparagine at position -1, an hydrophobic residue at position $x_2$ and negatively charged residues at positions +3 and +5), were found in fragments from various substrates. The simultaneous presence or absence of negatively charged residues at location +3 and +5 was a recurring pattern; we hypothesized that they correspond to the simultaneous presence or absence of stabilizing salt bridges with the positively charged lysines/arginines of Cn at locations 100,318 and/or 332.

Altogether, the sequence model recapitulated well-known biochemical features of the Cn interactions. We next used the sequence model to generate two libraries of candidate peptides, respectively using regular Monte Carlo sampling and so-called low-temperature sampling to

focus samples around with higher probability values, following Russ et al. [26]. The former peptides spanned a larger portion of the sequence space and were on average further away from the set of natural sequences, while the latter had higher probability scores (**S3E Fig**). After redundancy reduction, 180 regular samples and 361 low-temperature samples were selected for further analysis. For comparison purpose, we further included: (i) *As positive controls*, two synthetic peptides (PVIVIT and PKIVIT) known to bind Cn at high affinity, (ii) *As negative controls*, 36 purely random 16-length peptides (iii) *Natural binding fragments* from the four human NFATc1-4 genes, plus 70 additional representatives (**Methods**) of the MFA; the majority of these are expected to bind Cn. (iv) *As a baseline generative model*, 72 samples from the PSSM model. The full list of 784 sequences is reported in S4 Data.

## Library refinement by molecular docking and microarray binding assay (Step 3)

The sequence model a priori treats all natural sequences equally. However, their binding affinities span almost three orders of magnitude (0.5–250 uM). To further refine the list of candidate peptides, we estimated the docking energy score (where a lower score is better) using five available crystal structures of Cn bound to a PxIxIT-containing peptide and an ad-hoc template-based molecular docking followed by a flexible-backbone refinement pipeline based on Modeller [55] and PepCrawler [56] (**Fig 4A**, **Methods**). The docking energies were consistent from one Cn crystal structure to the other (**S4A Fig**), and correlated with the likelihood of the SGM (**S4B Fig**, Pearson correlation r = -0.21, $p < 10^{-8}$). Random or PSSM-designed sequences had significantly higher energy than natural or cRBM-designed ones ($p < 10^{-12}$, two-sided Mann-Whitney-Wilcoxon test). In contrast, there was no statistically significant difference between natural fragments and cRBM designs. However, the distribution of energies for the random peptides (negative controls) and natural binding peptides (positive controls) overlapped significantly: 14% of random peptides had lower energy scores than at least half of the natural peptides (**Fig 4B**). Hence, designing peptides solely based on docking would have likely led to a large number of false positives. To rationalize the docking energy score from the peptide sequence, we fitted an additive single-site model to the docking results by sparse linear regression (LASSO). The single-site model approximated well the docking energies results (cross-validation Pearson correlation r = 0.89, **S4E Fig**). Visualization of the regression coefficients (**Fig 4C**) confirmed that the motif sites were the most important ones, with highly hydrophobic side-chains (I,V,L,M,F) favored at these locations, as well as Proline at position "P". Flanking residues were overall less important but also contributed, in particular at position +3, where negatively charged residues (D,E) were favored, consistently with the sequence model. To evaluate the ability of the docking energy to discriminate between natural binders, we calculated approximate docking scores for all natural fragments using the single-site model and computed a per-substrate average (**Fig 4D**). The substrate with the lowest average docking energy was the A-kinase anchoring protein 79 (AKAP79), which is consistent with previous binding affinity measurements ($k_d$ ~0.1–2.0 uM) and its biological function. Indeed, AKAP79 tethers Cn to neural membranes next to synaptic clefts for phosphoregulation of synaptic signaling [59]. NFAT lies in the middle of the spectrum, consistent with previous observation that its affinity is fine-tuned to an intermediate value that prevents over-activation of NFAT in the absence of calcium [22].

Altogether, we conclude that the docking score can efficiently complement the evolutionary score by differentiating between natural genes with variable binding strengths. On the other hand, peptide design based solely on the docking protocol would have resulted in a highly hydrophobic binding motif, presumably with low solubility and high reactivity, as well as limited accuracy for flanking residues.

In parallel, selected peptides were tested for Cn binding on a chip microarray (PEPper-PRINT). 786 peptides were printed on the chip and were incubated with GST-tagged Cn overnight at 4˚C. Following extensive washing, binding was detected by applying a fluorescently labeled (Alexa-Fluor 647) GST antibody. After additional washing and drying, the microarray slide was scanned (**Fig 4E**), and fluorescent spots revealed peptides that bind Cn. The experiment was repeated five times, and Z-normalized fluorescence levels were determined following post-processing of the raw data (**Methods**, **S5A**, **S5B and S5C** **Fig**).

Although no statistically significant correlation was found between the experimentally determined fluorescence levels and either the sequence model or the docking scores, the positive outliers in the chip also had good sequence model and docking scores (**Figs 4F and S5D**). In addition, a false negative readout was observed, as the PVIVIT and PKIVIT peptides had only weak fluorescence levels and fluorescence was not well explained a-posteriori by a sequence-based single-site model (**S5E Fig**). We concluded that the experimental fluorescence level alone did not consistently indicate binding, but that positive hits could be considered reliable. Based on the analysis of the sequence likelihood scores, docking scores and fluorescence levels on the chip, 15 peptides were selected for further analysis: the PVIVIT peptide, four native fragments and ten cRBM-designed peptides.

## In-vitro quantitative binding assay and analysis (Step 4)

Selected peptides were synthesized and their ability to specifically bind the Cn PxIxIT binding site was evaluated by FP competition assay (**Methods**). Variable concentrations of each peptide were incubated in a solution of Cn complexed with fluorescently-labeled PVIVIT peptide, and FP levels indicated the peptide's ability to compete with the PVIVIT were read (**Fig 5**). After fitting the polarization values to a single site inhibition model, the corresponding IC50 values were extracted and reported in **Table 1**. We found that 7/10 synthetic peptides and 3/4 natural peptides successfully competed with PVIVIT for Cn binding, with IC50 values ranging from 1.17μM to 250μM. For comparison, we also tested non-labeled PVIVIT for self-competition using identical protocol, and found an IC50 of 10.2μM.

The best peptide, a fragment of the human C16Orf74 gene selected for its high sequence score, had an IC50 = 1.17 uM. This was consistent with its high gene-averaged docking score (rank 5/67, **Fig 4D**), and with recent binding affinity measurements for the CABIN1 gene, a close homolog (ATKFPPEITVTPPTPT) [60]. C16Orf74 features a highly hydrophobic PxIxIT-like motif, and, interestingly, a C-terminal proline-rich motif. Structural modeling with AlphaFold-multimer of C16Orf74 and PVIVIT suggests that the proline-rich adopts a rigid polyproline helical conformation that effectively extends the beta sheet, enabling additional interaction surface with Cn without any entropic cost (**S6 Fig**). We stress that such polyproline motif is extremely unlikely to be discovered by mutagenesis, as three simultaneous mutations to proline are required for the rigid structure to emerge. The second best peptide (IC50 = 17.5uM) was a fragment of the human AKAP79 gene, in agreement with its high sequence score, gene-averaged docking score (rank 1/67, **Fig 4D**) and previous studies.

Our best synthetic peptide, ADEAIPEIVISKPEEP (rbmTRESK hereafter), was obtained by low-temperature sampling of the cRBM and bound Cn with comparable strength as PVIVIT (IC50 = 14uM). It featured six mutations from its closest natural counterpart, the Cn-binding fragment of human TRESK protein and its IC50 was almost four times lower (IC50 = 54uM). A sequence with such a large number of mutations would have been difficult to reach via classical computational mutagenesis approach and almost impossible via experimental approach alone within a single screening round. Instead, rbmTRESK was effectively obtained by rational recombination of the left flanking residues of TRESK (**ADEAI**PQIVIDAGADE), the motif

**Table 1. List of peptide sequences characterized by competitive fluorescence polarization assay.** Abbreviations: IC50: half maximal inhibitory concentration; NB: No binding; low T: low temperature sampling.

| Name | Sequence | IC50 (uM) | Source | Closest natural sequence | Closest Organism | Closest Gene | # Mutations to closest natural sequence |
|---|---|---|---|---|---|---|---|
| C16Orf74 | KHLDVPDIIITPPTPT | 1.17 | Natural | KHLDVPDIIITPPTPT | Homo sapiens | C16Orf74 | 0 |
| PVIVIT | MAGPHPVIVITGPHEE | 10.2 | Positive control | / | / | / | / |
| rbmTRESK | ADEAIPEIVISKPEEP | 14 | Designed (low T) | ADEAVPQIIISAEELP | Homo sapiens | TRESK | 6 |
| AKAP79 | KRMEPIAIIITDTEIS | 17.5 | Natural | KRMEPIAIIITDTEIS | Homo sapiens | AKAP79 | 0 |
| TRESK | ADEAVPQIIISAEELP | 54 | Natural | ADEAVPQIIISAEELP | Homo sapiens | TRESK | 0 |
| rbmAKAP79 | AAGAGVGIVITVTEAE | 57 | Designed (low T) | AAGAGVGIVITVTEAE | Pelecanus crispus | AKAP79 | 2 |
| rbmTRESK_2 | ADEAIPEITITSAELP | 60 | Designed (low T) | ADEAIPQITITAEELP | Propithecus coquereli | TRESK | 3 |
| rbmAKAP79_2 | ADGAGVGIVITVTEAE | 69 | Designed (low T) | ADGAGVGIVITVTEAE | Pelecanus crispus | AKAP79 | 2 |
| rbmRIPOR2 | ASVSNPEITVTSAETE | 79 | Designed | QSQSNPEITVTPPETE | Austrofundulus limnaeus | RIPOR2 | 4 |
| rbmRIPOR2_2 | HVSSSPRITITPTQHR | 200 | Designed (low T) | HVSSSPDITATPTQHR | Fulmarus glacialis | RIPOR2 | 2 |
| rbmCAPN11 | TYWTNPQFRVTLEDPD | 257 | Designed (low T) | TYWTNPQFRITVEDPD | Schistosoma mansoni | CAPN11 | 2 |
| CAPN11 | TTAMNPQFFVQIPRTA | NB | Natural | TTAMNPQFFVQIPRTA | Daphnia magna | CAPN11 | 0 |
| rbmCAPN11_2 | TFHTNPQYRIKLEEPD | NB | Designed (low T) | TFHLNPQYRIKLEDPD | Hydra vulgaris | CAPN11 | 2 |
| rbmCAPN11_3 | TFWTNPQYRIKLEEAD | NB | Designed (low T) | TFWTNPQYLIKLEEED | Lonchura striata domestica | CAPN11 | 2 |
| rbmSFB3 | RTYINPFMLFEDMGSR | NB | Designed | RTYINPFMTFRSGGSR | Capronia coronata | SFB3 | 4 |

residues of KCNN3 (PTQNP**PEIVIS**SK**E**DS) and the right flanking residues of CAPN11 (TFWTNPQFKIYL**PEE**D).

Interestingly, the above peptides all featured a PxIxIT-like motif, but this was not necessary: peptides rbmAKAP79 and rbmAKAP79_2, both similar to the AKAP79 gene of Pelicanus crispus, successfully competed with PVIVIT binding despite lacking proline residues.

Finally, we found that all four natural or synthetic peptides similar to the CAPN11 gene consistently failed to compete with PVIVIT. There are three possible explanations: (i) they bind a distinct region of Cn, (ii) binding is dependent on post-translational modifications such as phosphorylation, (iii) they are bona fide non-binders, *i.e.* our training dataset was corrupted. In all cases, it would be straightforward to remove all CAPN11-like peptides from the multiple fragment alignment for a future screening round.

## Discussion

It is estimated that over half a million PPIs occur in the cell, among which many play important physiological roles and are potential therapeutic targets [1]. However, discovery of PPI modulators, especially in the form of small organic molecules, is hampered by the inherent physio-chemical properties of PPI interfaces, limited availability of structural data and accuracy of docking tools. This often prompted the conclusion that PPIs are "undruggable" targets [61]. Synthetic peptides capable of binding a target protein with high affinity and specificity, and interfering with its native protein-protein interactions are attractive reagents for basic systems biology research, structural characterization of protein-protein interactions, and drug

discovery campaigns. However, rational peptide design remains a major challenge, owing to the large search space, difficulty to estimate at high throughput the binding affinity and specificity in vitro or in silico, and necessity to integrate multiple design constraints.

Here, we proposed a novel integrative approach to design peptides targeting a specific binding site of a protein, based on protein fragments extracted from native interaction partners. After identification of native partners together with their interacting fragments using available experimental data and homology search, a sequence generative model (here, a cRBM) was trained and sampled from, yielding an in-silico library of $10^{3-4}$ "reversed-engineered" peptides. Peptides were subsequently filtered by a cost-effective and medium-throughput approach (template-based docking and microarray binding assay). Finally, a focused list of $\sim 10^1$ peptides was prioritized and their ability to interfere with the target protein-protein interaction(s) was quantified by FP assay. We applied our protocol to the Cn-PxIxIT complex, a key component of calcium signaling pathways, and found that 7/10 designed peptides and 3/4 natural peptides successfully bound the target binding site. Our most successful peptides were a previously overlooked natural peptide featuring a C-terminal proline-rich motif, and a designed recombinant peptide harboring six mutations from its closest natural counterpart. Both would be virtually impossible to reach via classical mutagenesis in a single screening round. These peptides set the basis for further optimization via in-vitro mutagenesis [62] or transfer learning [27], and/or multicomponent constructs [63] that also target the secondary LxVP binding site.

How robust is the protocol with respect to the choice of the sequence generative model? Here, we used compositional Restricted Boltzmann Machines (cRBM), an expressive generative model with a closed-form likelihood function and tractable number of parameters that generalizes over PSSM (i.e., independent model) and Potts models by incorporating sparse, high-order interaction terms. A protocol solely based on PSSMs, which neglects inter-dependencies between positions would yield a much lower success rate, as evidenced by (i) entropy calculations, which show that explicitly modeling inter-dependencies reduces the size of the space of candidate sequences by a factor of at least ~4000 (Methods) (ii) flexible docking simulations, which show that PSSM-generated sequences have worse docking energy than cRBM-generated ones on average and (iii) the microarray experiment, which did not identify any PSSM-generated sequence with high binding affinity to Cn. On the other hand, generative Potts models [64,26,65], which also integrate inter-dependencies, model equally well the distribution of natural binders (**Methods**, **S3A Fig**), and would likely perform comparably. Other machine learning generative models that account for inter-dependencies (see Wu. et al. [28] for an overview) may be suitable as well. On the other hand, it is unclear how well language model-based sequence generation protocols (see *e.g.*, [66,67]) would perform, as they are usually trained and evaluated on full-length sequences rather than on fragments; here, they may lack key contextual information (e.g., that the fragments belong to disordered regions).

Other important factors of success were (i) the availability of sufficient sequence and structural data about Cn substrates based on previous experiments (ii) the robustness of the SGM protocol with respect to corrupt training data and (iii) the complementarity between evolutionary-based and docking-based approaches. Indeed, while evolutionary-based models alone could not discriminate transient from tight natural binders, structure-based docking alone could not explain amino acid preferences for flanking residues and favored promiscuous, hydrophobic side-chains for central residues. Accordingly, we found that our cRBM model better predicted binding affinity changes upon mutation that ProteinMPNN [58], a recently published structure-based autoregressive graph neural network for sequence design. Synergy between evolutionary-based and structure-based protein design approaches is well-established [68–71] and therefore, although structure-based computational design methods are rapidly

improving [58,72,73], we expect that evolutionary information will still prove valuable in the future.

Our protocol has the following limitations:

i) Training the SGM requires a sufficiently diverse set of sequences. Thus, the protocol is only applicable if the interaction is highly conserved throughout evolution, and/or if multiple natural binders have been characterized. Although the protocol was only tested for the highly multivalent and thoroughly-studied Cn, we remark that many other protein targets of interest enjoy similar feats.

ii) There is no guarantee that all sequences in the multiple sequence alignment used for training bind the target. Indeed, pairing of interacting orthologs is a challenging problem that can only be approximately solved. Moreover, post-translational modifications frequently modulate binding of SLiMs.

iii) At least one experimental structure or reliable model must be available for at least one binder in order to perform template-based docking and scoring.

iv) In the microchip qualitative binding assay, peptides are chemically linked to the chip rather than freely moving in a solution phase. This may not accurately reflect their physical behavior in solution, hamper binding, and in turn, lead to high false negative rates. Here, we used quintuplicate to limit this issue.

## Materials and methods

### Construction of the multiple fragment alignment of natural binders

The following abbreviations are used: Catalytic domain of Cn: CnA, Regulatory domain of Cn: CnB, substrate protein: SP. Short Linear Motif: SLiM. A flow chart summarizing the protocol is shown in **S1A Fig**. The protocol initiates from a list of 67 (38 from yeast, 29 from human) experimentally validated substrate proteins of the two-domain Cn enzyme previously characterized in the literature. For each SP, the location of the fragment binding the Cn PxIxIT binding site was previously identified by phage display screening and/or SLiM matching [44,45]. Since generative modeling generally benefits from increased sequence diversity, we sought to augment this initial set by homology search. However, naive homology search was not suitable because Cn-SP interactions are not systematically conserved throughout evolution [44]; we instead proceeded as follows. For CnA, CnB and for each SP, initial seed alignments of interacting orthologs were first constructed. Orthologs were collected from the Homologene database if available, or via a BLAST search over UniProt (default parameters, top-100 hits, keeping only sequences with identical or synonymous gene name). For ordered proteins, the seed sequences were aligned using MAFFT [74] (default parameters). For disordered proteins (identified as such by IUPred2 [75]), visual inspection of MAFFT outputs revealed unsatisfactory alignments that did not consistently align the binding SLiMs. Instead, KMAD [76], a multiple alignment software tailored for disordered proteins was used (parameters: custom pattern "P.I.I.", add score = 100, substract score = 5, default otherwise). Next, for each Cn domain and SP, additional homologs were searched over the UniClust30 database (time stamp: 2018/06) using HHblits 3 (4 iterations, default values of other parameters) [77]. Non-interacting homologs were next filtered out using a variant [78] of the MirrorTree [79] approach. The intuition behind MirrorTree is that when two protein families interact, their respective phylogenetic trees tend to be similar. More specifically, if the interaction between protein 1 and 2 is conserved in species A and B, then their sequences should have diverged at a similar rate from one another $\mathrm{Sim}(P_1^A, P_1^B) \propto \mathrm{Sim}(P_2^A, P_2^G)$. Conversely, deviations from this pattern indicate possible gene duplication events that do not necessarily preserve functional interaction. Formally, there is for each SP a set of seed triplets [(CnA$^{\mathrm{SeedOrg1}}$, CnB$^{\mathrm{SeedOrg1}}$,

SP$^{SeedOrg1}$), (CnA$^{SeedOrg2}$, CnB$^{SeedOrg2}$, SP$^{SeedOrg2}$), ..] and a set of candidate triplets [(CnA$^{Org1}$, CnB$^{Org1}$, SP$^{Org1}$), (CnA$^{Org2}$, CnB$^{Org2}$, SP$^{Org2}$), ..] to be filtered out—in general there are multiple candidate triplets per organism. For each triplet $l$ and seed triplet $l'$ the sequence identity of each partner is computed: $S_{l,l',k} = \text{Sim}(P_k^l, P_k^{l'})$, where $k \in (CnA, CnB, SP)$ is the complex component index. Next, the Pearson correlation matrix is determined, $R_{l,k,k'} = \text{PearsonR}'_l(S_{l,l',k})$ followed by its off-diagonal average: $R_l = 1/3(R_{l,CnA,CnB} + R_{l,CnA,SP} + R_{l,CnB,SP})$. $R$ quantifies the overall consistency of the evolutionary divergence of the proposed triplet with respect to the seed triplets. Next, a single copy of CnA and CnB was identified for each species by maximizing the average $R_l$. Triplets involving another CnA/CnB copy were discarded, and among the remaining ones all triplets with $R_l$ above a threshold were kept. The threshold was determined individually for each SP, as the minimum over seed triplets of $R_l$ (*i.e.*, such that all seed triplets were kept).

Next, for each interacting SP, its presumed interacting fragment was extracted by taking all amino acids (including insertions) between columns i*-10 and I*+16, where i* is the starting location of the SLIMS for the seed sequences. The fragments were pooled together, and realigned with MAFFT (gap opening penalty: 6, gap extension penalty: 2). At this stage, visual inspection revealed fragments that clearly deviated from the main distribution (abnormally long, no visible SLiM), presumably because of sequence alignment errors or not being properly filtered out. To remove these sequences, a Restricted Boltzmann Machines with 10 dReLU hidden units and sparse regularization penalty $\lambda_1^2 = 10^{-2}$ (see next section) was trained, and the log-likelihood log$P$ was computed for each sequence. The distribution, shown in **S1B Fig** featured a heavy left tail, meaning that many sequences were outliers, belonging to sparsely populated regions of the sequence space. To determine a cut-off, the sequences were grouped by likelihood interval (dotted lines), and a sequence profile was computed for each group (**S1C Fig**). Sequences with Z-normalized likelihood score below -0.3 did not feature the expected SLiM motif, nor any significant sequence conservation, and were therefore discarded. After filtering, realignment and retraining, the new likelihood distribution (not shown) featured a unimodal shape consistent with previously studied models of protein families, and the sequence profile featured the expected conservation patterns. Finally, only 5 flanking residues were retained on each side to facilitate comparison with previous works. Note that we did not use the CnA/CnB alignments, as the binding site to PxIxIT was highly conserved and there was no coevolution between Cn and its substrates.

## Sequence generative modeling

cRBMs were trained on the multiple fragment alignment by Persistent Contrastive Divergence following [53] (source code available from https://github.com/jertubiana/PGM), and the following parameters were used: number of hidden units: from 5 to 30; hidden unit potential: double Rectified Linear Units (dReLU); batch size: 100; MCMC sampler: alternate Gibbs; number of Markov chains: 100; number of Monte Carlo steps between each gradient evaluation: 20; number of gradient updates: 20000; optimizer: ADAM with initial learning rate: $10^{-3}$, exponentially decaying after 50% of the training to $10^{-5}$, $\beta_1 = 0$, $\beta_2 = 0.99$, $\epsilon = 10^{-3}$. For the regularization, we used a sparse $L_1^2$ penalty on the weights (of strength $\lambda_1^2$ ranging from 0.0 to 1.0) and a $L_2$ penalty on the fields (of strength $10^{-2} \times \lambda_1^2$). Training samples were assigned a weight inversely proportional to the number of similar sequences in MSA with at most 1 similar amino acid. To calculate likelihood scores, the partition functions were evaluated using the Annealed importance Sampling algorithm, using $10^4$ intermediate temperatures and 10 repeats. To quantify the sparsity of the learnt sequence motifs, the fraction of non-zero weights was estimated through participation ratios as described in Eqs 20,21 of [53]. For parameter

selection, the MFA was split into training and validation sets such that sequences from training and validation differed by at least three residues. This was done by performing hierarchical clustering with single linkage merging criterion (scipy.cluster.hierarchy.single command), cutting the tree at 2 and assigning 80% of the clusters to train and 20% to validation. A grid search was performed over the number of hidden units and the regularization strength, and the model sparsity and held-out average log-likelihood were monitored (**S3A and S3B Fig**); the model that best compromise between good sparsity and likelihood was selected, corresponding to 30 hidden units and 0.25 $\lambda_1^2$ regularization strength. Its per-site likelihood was substantially better than the best independent/PSSM model (optimized over pseudo-count value) and slightly lower than the best Potts model trained using the same algorithm (optimized over $L_2$ regularization strength). After parameter selection, the best cRBM was retrained over the full MFA. After training, mutational landscapes shown in **Fig 3C and 3D** were computed by repeated application of Eq (1) for wild-type and single-point variants. The effective epistasis matrix of **S3D Fig** was computed via Eqs 15,16 of [53]. Approximately 10 000 peptides were sampled for regular and low-temperature sampling, and hierarchical clustering was subsequently performed to reduce it to 361 and 180 peptides (where the cluster representative is chosen as the sequence with highest cRBM likelihood). The MFA was similarly clustered to obtain the 70 representative fragments. Based on entropy calculations of the 1) PSSM distribution, 2) low-temperature PSSM distribution, 3) cRBM distribution and 4) low-temperature cRBM distribution, the size of the set of Cn-binding peptides was estimated to be respectively 1) $10^{17.3}$, 2) $10^{12.9}$, 3) $10^{13.7}$ and 4) $10^{2.8}$—a tiny fraction of the $10^{20.8}$ possible peptides of length 16.

## Template-based docking and binding scoring

A physical binding score was determined for each of the 768 candidate peptides by template-based docking as follows. Five structures of Cn catalytic domain in complex with various PxIxIT-containing peptides were collected from the pdb: 2p6b (PVIVIT), 3ll8 (AKAP79), 6uuq (RCAN1), 6nuf (NHE1) and 2jog (PVIVIT, NMR) [51,80–83]. Given a peptide sequence and template complex, the candidate and template peptide sequences were aligned (by motif matching), then 100 homology structural models for the peptide were built using Modeller [55]. The candidate peptide was superimposed onto the template peptide and translated away from Cn if steric clashes occurred (<2A center-center distance between any pair of atoms). Then, a model in extended conformation and forming many contacts was selected by maximizing the cost function Coverage + 0.05 * NumAtomContacts + 0.05 * Extension where NumAtomContacts is the number of atomic contacts (heavy atoms only, 4A distance cutoff between atom centers), Coverage is the number of peptide residues forming at least one atomic contact and Extension is the euclidean distance between the C-terminal and N-terminal Calpha atoms. Next, the initial conformation was refined using PepCrawler [56], a fast, flexible backbone conformational sampling algorithm based on rapidly-exploring random trees and an all-atom energy function. The above homology modeling + refinement protocol was repeated 10 times for each structure and the configuration with minimum energy was retained. In total, each peptide was docked 50 times, corresponding to ~1–2 days of computation on a single CPU core of an Intel Xeon Phi processor. The funnel score—a measure of the steepness of the energy landscape around the minimal energy configuration—was also computed, as it was previously shown to characterize good peptide inhibitors [12,56]. **S4A Fig** (resp. **b**) shows the cross-correlation matrix between binding (resp. funnel) scores determined from the five structures. The docking energies correlated well (r~0.5 for all pairs) but not the funnel scores, allegedly due to the homology modeling step or to the relatively long peptide length. Thus, only the binder energy score was retained.

### Peptide array readout and analysis

The peptide array was prepared as a custom array pepper chip (PEPperPRINT). For binding detection, 1mg/ml GST-tagged Cn was incubated overnight on the microarray at 4˚C. Following extensive washing, fluorescently labeled (Alexa-Fluor 647) GST-antibody. After additional washing and drying according to suggested manufacturer protocol, the microarray slide was scanned with an InnoScan 1100 scanner. The experiment was repeated five times. Each scan was analyzed as follows: a grid was overlaid using the border HA markers to determine regions of interest (ROI) for each peptide (two ROIs per peptide) and the logarithm of the average fluorescence intensity was computed for each ROI. The baseline fluorescence level was not uniform throughout the array as evidenced from scatter plots of log-fluorescence intensity against row and column index (**S5A and S5B Fig**). To remove spatial artifacts, a position-dependent baseline fluorescence was fitted using a second-order polynomial, and subtracted to the fluorescence. Fluorescence levels were next averaged over the two ROI for each peptide and Z-normalized. The peptides lying along the border were found to have significantly higher fluorescence level due to oversplash from the border HA markers (**S5C Fig**) and were not further analyzed. We checked that there was no significant oversplash within the interior of the chip by monitoring the spatial autocorrelation function of fluorescence scores. The Z-scores were averaged over the five repetitions to yield one fluorescence score per peptide.

### Cn expression and purification

The catalytic domain of Cn was expressed as described in [84,85]. Briefly, the gene encoding to residues 2–347 with substitutions Y341S, L343A, and M347D was expressed as a cleavable GST fusion protein in *E. coli* BL21 cells. Growing in LB, after reaching to OD(600nm) = 0.8, protein expression was induced by the addition of 1 mM IPTG at 25˚C. The cells were harvested after 16h and resuspended in PBS-based lysis buffer suitable for downstream purification onto GST column (Glutathione Sepharose 4 Fast Flow) of the soluble fraction after disrupted by sonication and remove of all non-soluble debris by centrifuge. Elution from the GST column was further purified by size exclusion chromatography with the superdex75 column.

### Peptide synthesis

Peptides were synthesized with N-ter acetylation and C-ter amidation by peptide2 Inc (USA).

### Fluorescence polarization experiment

Fluorescence measurements were performed on samples arrayed in a 96-well plate using Bio-tek HybridH1 reader equipped with a polarized optic system. All measurements were done in triplicate. Competition was evaluated by adding variable concentrations of each non-labeled tested peptide to wells containing 100nM of FITC-labeled PVIVIT peptide and 4uM of Cn A. Experimental polarization data from simple and competitive binding experiments were fitted using GraphPad Prism7, with error bars representing standard deviation.

### Post-hoc analysis

All statistical analysis, including statistical tests, LASSO regression (Used in Figs 4C and **S5E** for fitting single-site models to the results of the molecular docking and microarray binding experiment) and T-SNE dimensionality reduction (for visualizing the distribution of natural fragments and highlighting clusters in **S2A** and **S2B Fig**) were performed using the numpy, scipy and scikit-learn Python packages.

## Supporting information

**S1 Table. Performance comparison of various structure-based and evolutionary-based models for predicting the effect of single-point mutations on binding affinity with Calcineurin for four peptides, see Fig 3C–3E.** Data from Nguyen et al. [24].
(XLSX)

**S1 Fig. Construction and refinement of the multiple fragment alignment (MFA).** (a) Overview of the protocol for constructing the multiple fragment alignment. (b,c) cRBM-based refinement of the MFA obtained from the homology search: A cRBM model is trained on the MFA and likelihood scores are computed for all sequences (higher is better). Sequences with low likelihood values do not share the main conservation and coevolution patterns of the others and may be discarded. To determine a cut-off, we group sequences by likelihood, and visualize the sequence profile of each subgroup (c). Sequences with Z< -0.3 do not feature any conservation pattern, and are considered outliers.
(PNG)

**S2 Fig.** Selected data visualization of the multiple fragment alignment. (a,b) T-SNE visualization of the MFA, colored by phyla/gene reveals that fragments mainly cluster by gene. (c) Gene-specific sequence profiles, revealing a diversity of conserved binding motifs. B denotes the number of unique fragments found.
(PNG)

**S3 Fig. Additional information on the sequence model.** (a,b) Model selection protocol. A grid search is performed over the number of hidden units and values of sparse penalty. The model that achieves the best compromise between accuracy (high likelihood) and interpretability (low motif sparsity) is selected (black triangle). (c) Distribution of likelihood changes upon mutation grouped by position. The distribution is computed over all 19 possible mutations at each position for 100 representative fragments from the alignment (randomly selected by Kmeans++ algorithm). Positions with low average values are less tolerant to mutations and presumably more important for functionality. (d) Effective epistatic couplings learnt by the model, indicating significant covariation between core and flanking residues. (e) The quality-diversity trade-off of generated sequences. Scatter plot of sequence likelihood against number of mutations to the closest natural fragment. Sequences with likelihood similar or higher to the one of natural sequences but distinct from them can be generated. Conversely, random or PSSM-generated sequences are further away but distinguishable from natural fragments.
(PNG)

**S4 Fig. Additional information regarding the structural modeling protocol.** (a,b) Pearson cross-correlation matrix of the docking energy and funnel scores between five repeats using different crystal structures for Cn. (c,d) Scatter plot of the docking energy and funnel scores averaged over the five repeats against the sequence model likelihood; both are weakly but significantly correlated. (e) Single-site model fit of the docking energy scores by LASSO regression: scatter plot between docking energies and cross-validated predictions.
(PNG)

**S5 Fig. Additional details regarding the Microassay experiment data processing.** (a,b) Scatter plot of the fluorescence intensity level against the row and column index along the chip for one of the seven repeats (one point per peptide). A trend is fitted (red curve) and removed before further analysis. (c) Distribution of fluorescence intensity level (logarithmic scale, normalized to zero mean and unit variance, averaged over the seven repeats), for peptides printed at the border and at the interior of the chip. Spillover from the neighboring fluorescent tags

results in higher fluorescence levels for border peptides; positive hits along the border were ignored in downstream analysis. (d) Scatter plot of the docking energy score against the fluorescence level. (e) Single-site model fit of the docking energy scores by LASSO regression: scatter plot between docking energies and cross-validated predictions.
(PNG)

**S6 Fig. Computational modeling of the Cn-PVIVIT and Cn-C16orf74 complexes using AlphaFold-multimer.** Calcineurin is shown in surface representation, colored by electrostatic potential. PVIVIT and C16orf74 peptides are shown in stick representation, respectively in green and orange. Visualization was performed using ChimeraX.
(PNG)

**S7 Fig. FP competition of peptides against fluorescently-labeled PVIVIT peptide.** Variable concentrations of each of the peptides were incubated with the co-complex of Cn and PVIVIT. Polarization values were fitted to a single site inhibition model and IC50 values were extracted.
(PNG)

**S1 Data. List of natural Calcineurin-binding fragments identified by previous studies.**
(XLSX)

**S2 Data. Multiple Fragment Alignment used for training the sequence generative model.**
(FASTA)

**S3 Data. Sequence motifs learned by the sequence generative model.**
(XLSX)

**S4 Data. List of selected peptides evaluated on the chip-based assay.**
(PDF)

**S5 Data. Linux binary executable for PepCrawler.**
(ZIP)

## Acknowledgments

J.T. acknowledges Dina Schneidman-Duhovny for helpful discussions and Naama Hurwitz for her help with PepCrawler.

## Author Contributions

**Conceptualization:** Jérôme Tubiana, Haim J. Wolfson, Maayan Gal.

**Data curation:** Jérôme Tubiana.

**Formal analysis:** Jérôme Tubiana.

**Funding acquisition:** Jérôme Tubiana, Haim J. Wolfson, Maayan Gal.

**Investigation:** Jérôme Tubiana, Lucia Adriana-Lifshits, Michael Nissan, Matan Gabay, Inbal Sher, Marina Sova, Maayan Gal.

**Methodology:** Jérôme Tubiana.

**Project administration:** Haim J. Wolfson, Maayan Gal.

**Software:** Jérôme Tubiana.

**Supervision:** Haim J. Wolfson, Maayan Gal.

**Validation:** Jérôme Tubiana, Lucia Adriana-Lifshits, Michael Nissan, Matan Gabay, Inbal Sher, Marina Sova, Maayan Gal.

**Visualization:** Jérôme Tubiana, Maayan Gal.

**Writing – original draft:** Jérôme Tubiana, Maayan Gal.

**Writing – review & editing:** Jérôme Tubiana, Lucia Adriana-Lifshits, Michael Nissan, Matan Gabay, Inbal Sher, Marina Sova, Haim J. Wolfson, Maayan Gal.

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
