## [Decision Letter · Decision Letter 0]

7 Dec 2022

Dear Dr. Tubiana,

Thank you very much for submitting your manuscript "Funneling modulatory peptide design with generative models: discovery and characterization of disruptors of calcineurin protein-protein interactions" for consideration at PLOS Computational Biology. As with all papers reviewed by the journal, your manuscript was reviewed by members of the editorial board and by several independent reviewers. The reviewers appreciated the attention to an important topic. The two external reviewers  found the work interesting and robust. Based on the reviews, we are likely to accept this manuscript for publication, providing that you modify the manuscript according to the review recommendations.

Sincerely,

Elena Papaleo, PhD

Academic Editor

PLOS Computational Biology

Arne Elofsson

Section Editor

PLOS Computational Biology

The manuscript has been carefully evaluated by two external reviewers who found the work interesting and robust but had minor comments that need to be taken into account before recommending the article for publication.

Reviewer's Responses to Questions

**Comments to the Authors:**

Reviewer #1: The authors introduce an interesting methodology for computer-aided SLiM design, making use of interesting machine learning methods and experimental validation of suggested targets. While some of the in vitro and in silico methods did not yield the expected results in regards to controls, this should not preclude the publication of this novel approach to small peptide design for protein-protein binding.

As for revision requests, two minor issues stand out and, in my opinion, should be corrected before publication.

Line 246 - The trade-off between accuracy and interpretability, present in Supplementary figure 3(a,b) should be added to the main text and be expanded upon.

Line 292-295 - Please be more explicit in regards to which positive and negative controls were selected and how they were selected, as it is especially important considering the lack of control correlation later observed in silico and in vitro.

Reviewer #2: I have uploaded my comments.

**Have the authors made all data and (if applicable) computational code underlying the findings in their manuscript fully available?**

Reviewer #1: Yes

Reviewer #2: None

PLOS authors have the option to publish the peer review history of their article (what does this mean?). If published, this will include your full peer review and any attached files.

Reviewer #1: **Yes: **João M. Martins

Reviewer #2: No

Figure Files:

Data Requirements:

Reproducibility:

References:

---

## [Decision Letter · Decision Letter 1]

16 Jan 2023

Dear Dr. Tubiana,

We are pleased to inform you that your manuscript 'Funneling modulatory peptide design with generative models: discovery and characterization of disruptors of calcineurin protein-protein interactions' has been provisionally accepted for publication in PLOS Computational Biology.

Best regards,

Elena Papaleo, PhD

Academic Editor

PLOS Computational Biology

Arne Elofsson

Section Editor

PLOS Computational Biology

Reviewer's Responses to Questions

**Comments to the Authors:**

Reviewer #2: The authors responded reasonably to most questions and the article was improved.

I strongly recommend to consider the paper for publication.

**Have the authors made all data and (if applicable) computational code underlying the findings in their manuscript fully available?**

Reviewer #2: None

PLOS authors have the option to publish the peer review history of their article (what does this mean?). If published, this will include your full peer review and any attached files.

Reviewer #2: No

---

## [Editor Report · Acceptance letter]

30 Jan 2023

PCOMPBIOL-D-22-01508R1 

Funneling modulatory peptide design with generative models: discovery and characterization of disruptors of calcineurin protein-protein interactions

Dear Dr Tubiana,

I am pleased to inform you that your manuscript has been formally accepted for publication in PLOS Computational Biology. Your manuscript is now with our production department and you will be notified of the publication date in due course.

With kind regards,

Zsofi Zombor
